# Learning to Decompose Visual Features with Latent Textual Prompts

**Feng Wang[1], Manling Li[2], Xudong Lin[3], Hairong Lv[1], Alexander G. Schwing[2] & Heng Ji[2]**
[1]Tsinghua University    [2]University of Illinois at Urbana-Champaign    [3]Columbia University

## Abstract

Recent advances in pre-training vision-language models like CLIP (Radford et al., 2021) have shown great potential in learning transferable visual representations. Nonetheless, for downstream inference, CLIP-like models suffer from either 1) degraded accuracy and robustness when inferring by retrieving textual class names (the zero-shot protocol); or 2) breaking the well-established vision-language alignment (linear probing). To combine the best of both worlds, we propose **De**composed **F**eature Pr**o**mpting (DeFo). DeFo maintains the dual-model architecture yet leverages learnable embeddings as textual input and performs classification with an additional linear layer. As a result, we find DeFo to be able to extract decomposed visual features with the help of textual prompts and to allow a scalable size of language inputs. Our empirical study shows DeFo's significance in improving the vision-language models. For example, DeFo obtains 73.2% test accuracy on ImageNet with a ResNet-50 backbone without tuning any pretrained weights of both the vision and language encoder, outperforming zero-shot CLIP by a large margin of 15.0%, and outperforming state-of-the-art vision-language prompt tuning by 7.6%.

## 1 Introduction

Language-guided visual pretraining has gained a lot of attention and shows great promise in learning transferable image representations. By establishing a connection between images and natural language, recent vision-language models are able to turn visual inference over a restricted number of classes into zero-shot open-vocabulary inference (Radford et al., 2021; Jia et al., 2021; Pham et al., 2021).

One of the recent successes for zero-shot inference is the contrastive language-image pretraining (CLIP) model (Radford et al., 2021). It uses 400 million image-text pairs to learn an alignment between visual and textual representations obtained from a vision encoder and a language encoder respectively. In downstream applications, CLIP-like models (Radford et al., 2021; Jia et al., 2021; Pham et al., 2021) then perform zero-shot inference by **hard-target retrieval**, i.e., they directly compute the distance between a vectorial image representation obtained from the vision encoder, and representations of text prompts (e.g., "a photo of an airplane" or "a photo of an automobile") obtained from the language encoder. The target class (e.g., "airplane" or "automobile") corresponding to the text prompt with the smallest distance to the vector representing the image constitutes the zero-shot inference result. When annotations are given, simple linear probing (i.e., removing the language encoder, fine-tuning of the vision encoder and training of a classifier on top of the vision encoder) further improves the results (Radford et al., 2021). Moreover, context optimization (CoOp) (Zhou et al., 2021) replaces the hand-crafted prefix or suffix (e.g., "a photo of a") of the text prompts by trainable embedding vectors.

However, the zero-shot CLIP and CoOp infer using hard textual targets, i.e., the class names, which results in two main challenges. First, class names in text prompts (e.g., "airplane" or "automobile"), as used in zero-shot CLIP and CoOp inference, do not permit to accurately summarize the semantic information of an image. Therefore, inference is very sensitive to the words chosen for class names. We refer to this challenge as **expressive sensitivity**. Empirically, this challenge causes zero-shot CLIP and CoOp to struggle to achieve as competitive results as linear probing with the same image encoder when downstream training data is available (e.g., 58.2% accuracy vs. 72.3%

on ImageNet (Deng et al., 2009)). Moreover, this sensitivity can be observed by modifying class names. Fore example, for zero-shot inference on CIFAR-10 (Krizhevsky et al., 2009), CLIP obtains an accuracy of 63.7% when the original class names are used. Notably, simply replacing or extending the class names with suitable synonyms[1] (e.g., "plane" and "car" rather than "airplane" and "automobile") can improve accuracy to 79.6%, which highlights the challenge of expressive sensitivity.

Second, despite the fact that hundreds of millions of pretraining samples cover a large number of concepts that can possibly appear in downstream datasets, zero-shot inference continues to struggle to recognize rare objects. We refer to this as the **conceptual sensitivity**. For example, zero-shot CLIP is only 38.5% accurate when classifying EuroSAT satellite images (Helber et al., 2019), which is much lower than the result of a supervised ResNet-50 (He et al., 2016) encoder (93.4%). Also, zero-shot CLIP with a ResNet-50 encoder achieves less than 90% accuracy on MNIST (LeCun, 1998), which can even be outperformed by a simple logistic regression model. While linear probing is a straightforward way to improve results, removing of the language encoder breaks the vision-language alignment that is learned from the pretraining data, and therefore degrades few-shot and transfer learning performance.

In this paper, we propose **De**composed **F**eature Pr**o**mpting (DeFo), which turns the hard-target-retrieval paradigm of CLIP and CoOp into dual-model feature prompting. Specifically, DeFo 1) provides to the language encoder a set of learnable embedding sequences which are independent of the hard semantic targets; and 2) performs classification by tuning an additional layer. As a result, DeFo does not rely on the textual representations of class names being classification targets, which addresses the issues of **expressive sensitivity** and **conceptual sensitivity**. Meanwhile, DeFo maintains the dual-model architecture, which enables the model to leverage the language information, so that few-shot and transfer learning performance can be boosted.

DeFo results show the significance of addressing the sensitivity challenges of CLIP-like models. For example, with a ResNet-50 backbone, DeFo achieves 73.2% test accuracy on ImageNet without modifying any pretrained weight of the image and text encoders, outperforming vanilla CLIP by a large margin of 15.0% and outperforming CoOp by 7.6%. In a variety of visual contexts, DeFo attains an average accuracy of 79.9% over 11 image classification benchmarks, which is 21.0% higher than that of zero-shot CLIP and 6.2% higher than CoOp.

## 2 RELATED WORK

Pretraining-finetuning has long been a dominant paradigm of transfer learning in machine learning, computer vision, and natural language processing. Generally, pretraining a vision encoder by generative objectives (Bao et al., 2021; He et al., 2022) or discriminative objectives (He et al., 2020; Chen et al., 2020; Grill et al., 2020; Caron et al., 2021) at the scale of one to ten million images (Deng et al., 2009) is sufficient to yield good visual representations and strong predictive performance in downstream visual tasks. However, without the supervision from other modalities, such pretrained models require task-specific finetuning (Bao et al., 2021; He et al., 2022; O Pinheiro et al., 2020; Wang et al., 2022a; Lin et al., 2022a) or linear probing He et al. (2020); Chen et al. (2020) for reasonably domain-adapted predictions.

The contrastive language-image pretraining (CLIP) (Radford et al., 2021) method instead jointly pretrains a vision encoder and a text encoder on 400 million curated image-text pairs, with a contrastive objective (Gutmann & Hyvärinen, 2010) that matches the visual and textual representations. In downstream applications, CLIP achieves competitive results in various vision or vision-language tasks such as image classification (Zhou et al., 2021; Gao et al., 2021), dense prediction (Rao et al., 2022), video-language tasks (Luo et al., 2021; Lin et al., 2022b; Wang et al., 2022b), image manipulation (Patashnik et al., 2021), and multimedia event extraction (Li et al., 2022).

Following the success of CLIP, the ALIGN (Jia et al., 2021) model leverages a noisy dataset of 1.8 billion image-text pairs to scale up vision-language representation learning, and the BASIC (Pham et al., 2021) model further scales up this approach in terms of data and model size. Based on the success of CLIP-like vision-language pretraining, a series of follow-up inference approaches are proposed to improve classification results. For example, Zhou et al. (2021) propose CoOp to learn

---

[1]We use WordNet (Fellbaum, 2010) to find synonyms.

context information in downstream datasets, and Gao et al. (2021) propose CLIP-Adapter to learn domain-adaptation for vision-language models. Further, following CoOp, Zhou et al. (2022) propose CoCoOp to enhance the performance in unseen classes; and similarly, following CLIP-Adapter, Zhang et al. (2021) propose Tip-Adapter to explore non-parametric adaptation layers. Despite the progress these methods (Zhou et al., 2021; Gao et al., 2021; Zhou et al., 2022) have achieved in downstream predictive performance, they do not change CLIP's inference paradigm of retrieving class names. Hence, the challenges of expressive sensitivity and conceptual sensitivity remain.

## 3 METHODOLOGY

As shown in Figure 1, our DeFo follows the dual-model architecture of CLIP, i.e., we use a vision encoder and a language encoder which map the visual inputs and textual inputs into the same latent space. However, in DeFo, the language encoder plays a different role from that in the zero-shot CLIP. Specifically, CLIP directly constructs hard targets for classification by feeding the language encoder with $k$ textual queries (e.g., "a photo of cat", "a photo of dog", ...), where $k$ is the number of classes and each query corresponds to a specific one. As explained in Section 1, this inference protocol leads to **expressive sensitivity** and **conceptual sensitivity** challenges which incurs degradation of accuracy and robustness.

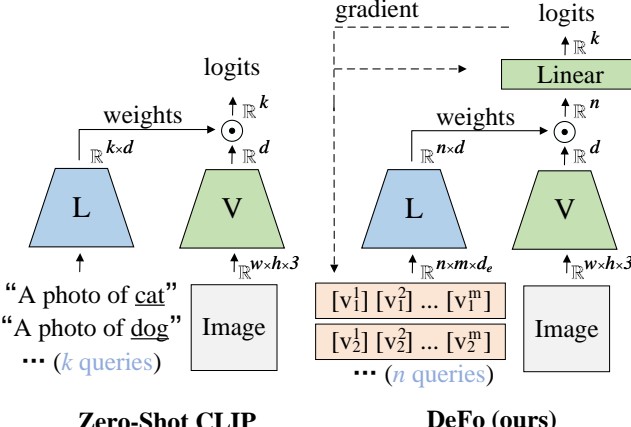

Figure 1: An architectural comparison between our DeFo and CLIP. "V" and "L" denotes vision and language encoder respectively and their weights are fixed. DeFo leverages sequences of trainable embedding vectors ($[v_i^j]$) as textual input and maps decomposed visual features by a linear layer.

In contrast, in DeFo, we change the existing paradigm of hard-target retrieval while maintaining the vision-language encoder architecture to learn decomposed visual features. Specifically, DeFo aims to utilize the language encoder to construct a projection matrix that maps the visual features from the $d$-dimensional CLIP latent space to a new $n$-dimensional feature space. To this end, we feed the language encoder with $n$ trainable text queries and then perform classification by an additional linear layer. By jointly tuning both the text queries and the classification layer, DeFo is able to learn textual prompts of detailed visual features and a robust feature mapping for classification.

Overall, DeFo has two main benefits compared to CLIP-like models. First, compared with hard-target-based inference protocols such as the zero-shot CLIP and CoOp (Zhou et al., 2021), DeFo removes the expressive and conceptual sensitivity challenges which significantly improves accuracy and robustness of downstream performance (see Table 1 and 4). Next, compared with linear probing which discards textual information, the optimization of the projection matrix in DeFo is bounded by the text encoder which results in the need for much fewer training samples to achieve good performance (see Table 2). Moreover, also note that in DeFo the number of textual queries $n$ is independent of the number of classes $k$, so the query size is scalable to fit specific downstream tasks. Next, we detail the DeFo and compare it to existing methods.

### 3.1 DUAL-MODEL INFERENCE

As shown in Figure 1, DeFo uses a visual encoder $g_V : \mathbb{R}^{w \times h \times 3} \to \mathbb{R}^d$ and a language encoder $g_L : \mathbb{R}^{m \times d_e} \to \mathbb{R}^d$ to extract image and text representations, respectively. For this, the visual inputs are 3-channel images of shape $w \times h$, and the language inputs are sentences with $m$ words where each word is embedded into a $d_e$-dimensional vector. Both the visual and textual features are then mapped into a $d$-dimensional latent space, i.e., we get an image representation vector $\boldsymbol{f}_I \in \mathbb{R}^d$ and $n$ text representation vectors $\boldsymbol{f}_T^1, \boldsymbol{f}_T^2, \ldots, \boldsymbol{f}_T^n \in \mathbb{R}^d$, where $n$ denotes the number of query sentences used for the encoder $g_L$. By applying the dot product between $\boldsymbol{f}_I$ and each of the $\boldsymbol{f}_T^i$ (note that both

$\boldsymbol{f}_I$ and $\boldsymbol{f}_T^i$ are $\ell_2$ normalized vectors, i.e., $\|\boldsymbol{f}_I\|_2 = \|\boldsymbol{f}_T^i\|_2 = 1$), we get an $n$-dimensional vector, where the $i$-th element measures the similarity between the image and the $i$-th text query.

CLIP and CoOp directly use this vector to predict the label of the image, because each text query in their settings corresponds to a specific class. Formally, CLIP and CoOp have $n = k$, where $k$ is the number of classes to be inferred, and the probability of the image belonging to the $i$-th class is computed by

$$p_i = \frac{\exp(\langle \boldsymbol{f}_I, \boldsymbol{f}_T^i \rangle / \tau)}{\sum_{j=1}^{k} \exp(\langle \boldsymbol{f}_I, \boldsymbol{f}_T^j \rangle / \tau)}, \tag{1}$$

where $\langle \cdot, \cdot \rangle$ denotes the dot product and $\tau$ is a temperature coefficient.

Instead, DeFo decouples the text queries from specific classes. Specifically, we use a scalable number of queries, i.e., the number $n$ is not limited to be equal to $k$, and perform classification by an additional linear layer that maps the $n$-dimensional feature vectors to $k$-dimensional vectors. The probabilities are then computed by the softmax of the $k$-dimensional vector. Note that only this linear classification layer and the textual queries are trainable in DeFo. We fix the weights of both the text encoder and the image encoder to maintain the vision-language alignment.

## 3.2 TRAINABLE TEXT EMBEDDINGS

The language encoder $g_L$ receives sequences of $d_e$-dimensional embedding vectors as input. When natural language is used, each word in the vocabulary first needs to be encoded into a $d_e$-dimensional embedding. In DeFo, we skip the process of designing hand-crafted prompts with natural language. Instead, we directly optimize the word embeddings via back-propagation. Specifically, we initialize $n$ independent sequences of text embeddings where each sequence consists of $m$ $d_e$-dimensional vectors in the form of "[v$^1$] [v$^2$] $\ldots$ [v$^m$]". The total "textual" input of DeFo can be written as a tensor $\mathbf{X}_L \in \mathbb{R}^{n \times m \times d_e}$. Note that here we assign the same length $m$ to each query for easy comprehension and implementation. In practice, the design of DeFo's input is more flexible and the length of each query is not required to be identical.

By optimizing $\mathbf{X}_L$, DeFo makes CLIP-like vision-language models free from both hand-crafted prompts and annotations such as class names. In this way we address the issues of expressive and conceptual sensitivity caused by using class names as hard targets.

## 3.3 COMPARISON TO EXISTING METHODS

As illustrated in Figure 1, zero-shot CLIP has no trainable parameters. The textual queries are composed by a hand-crafted prompt and class names that describe the semantic targets of the categories. The linear-probing CLIP uses only the vision encoder for classification. Without the assistance of textual representations, this method has to utilize an additional linear layer to map the visual features from the latent space ($d$-dimensional) to the output space ($k$-dimensional), which introduces $N = d \times k$ additionally trainable parameters. CoOp (Zhou et al., 2021) mostly follows the architecture of zero-shot CLIP, yet replaces CLIP's hand-crafted prompt by a sequence of trainable text embeddings, with $N = k \times m \times d_e$ learnable parameters for class-specific prompts.

Intuitively, both CoOp and our DeFo use trainable text embeddings as inputs of the language encoder. Both methods differ in that the number of textual queries $n$ is independent from the number of classes $k$ for DeFo, and the queries are not composed using class names. Therefore, DeFo has a scalable size of additionally learnable parameters. Specifically, it introduces in total $N = n \times (m \times d_e + k)$ trainable parameters, which scales linearly with the number of queries $n$. For example, with $n = 256$ and $m = 16$ in ImageNet ($k = 1000$), DeFo introduces 2.4M learnable parameters while attaining 72.3% accuracy, which outperforms CoOp (65.6%) who has 8.2M learnable parameters and CLIP-Adapter (63.6%) who has 1M learnable parameters.

In addition, compared to linear probing which directly maps the $d$-dimensional latent features to output logits, DeFo also uses a linear layer but maps $n$-dimensional features. In this way, DeFo is able to first project visual features with the assistance of $n$ textual representation vectors, which provides DeFo with significantly better few-shot performance and interpretability than linear probing.

Table 1: Test accuracy on ImageNet (%). Results with † are taken from Zhou et al. (2021), and those with ‡ are taken from Zhang et al. (2021). Our results are marked in  gray . The best results are **bolded**. The results without using text encoder are de-emphasized.

| Method | RN-50 | RN-101 | ViT-B/32 | ViT-B/16 |
|---|---|---|---|---|
| Zero-Shot CLIP (Radford et al., 2021) | 58.2 | 61.5 | 62.0 | 66.9 |
| Linear-Probing CLIP | 72.8 | **75.5** | 76.0 | 79.5 |
| Prompt Ensembling | 60.4$^{†}$ | 62.5$^{†}$ | 63.7$^{†}$ | 68.7$^{†}$ |
| CoOp (Zhou et al., 2021) | 65.6 | 67.8 | 68.0 | 72.4 |
| CoCoOp (Zhou et al., 2022) | 65.1 | 67.1 | - | - |
| Target Optimization (our ablation) | 71.4 | 73.2 | 74.0 | 78.1 |
| CLIP-Adapter (Gao et al., 2021) | 63.6$^{‡}$ | 65.4$^{‡}$ | 66.2$^{‡}$ | 71.1$^{‡}$ |
| Tip-Adapter (Zhang et al., 2021) | 62.0$^{‡}$ | 64.8$^{‡}$ | 65.6$^{‡}$ | 70.8$^{‡}$ |
| DeFo (ours) | **73.2** | **75.5** | **76.2** | **80.2** |

# 4 EXPERIMENTS

## 4.1 EXPERIMENTAL SETUP

### 4.1.1 BASELINE MODELS

DeFo is based on CLIP (Radford et al., 2021) for an easy comparison to the other baselines (Zhou et al., 2021; 2022; Gao et al., 2021). For CLIP, we mainly explore two inference protocols, zero-shot and linear probing. Zero-shot CLIP requires no extra training data and it infers by directly matching image representation to the text representation of class names with hand-crafted prompts. Linear-probing CLIP drops the text encoder and instead attaches a randomly initialized linear layer to the image encoder, and then tunes only this linear layer with downstream training data for domain-adapted classification.

CoOp (Zhou et al., 2021) and CLIP-Adapter (Gao et al., 2021) succeed in improving CLIP inference performance so they serve as the primary baselines to our DeFo. To give more comprehensive results, we also compare DeFo in ImageNet to the recent baselines of CoCoOp (Zhou et al., 2022) and Tip-Adapter (Zhang et al., 2021), which are direct extensions for CoOp (Zhou et al., 2021) and CLIP-Adapter (Gao et al., 2021). Note that we do not expect CoCoOp and Tip-Adapter to yield better results than their base models CoOp and CLIP-Adapter because they are proposed to address a different problem (discussed in Section 2). We report the results of Tip-Adapter without its further fine-tuning (Zhang et al., 2021) and all the baselines follow the pre-processing of CoOp for a fair comparison. Further, in this paper, we develop another baseline called "**Target Optimization**", which uses learnable embedding vectors as class names combined with a hand-crafted prompt prefix. Target Optimization can be regarded as an ablated version of DeFo, which helps to understand the importance of the learnable embeddings.

### 4.1.2 DATASETS

We follow prior methods to select 11 publicly available datasets, i.e., ImageNet (Deng et al., 2009), Food101 (Bossard et al., 2014), OxfordPets (Parkhi et al., 2012), Caltech101 (Fei-Fei et al., 2004), SUN397 (Xiao et al., 2010), UCF101 (Soomro et al., 2012), StanfordCars (Krause et al., 2013), FGVCAircraft (Maji et al., 2013), DTD (Cimpoi et al., 2014), Flowers102 (Nilsback & Zisserman, 2008), and EuroSAT (Helber et al., 2019). The categories in these 11 datasets include natural objects, scenes, human actions and fine-grained features such as textures and satellite imagery, which could cover general semantic targets of visual understanding tasks.

For the domain-generalization study, we also evaluate the models on four ImageNet-variant datasets, namely, ImageNet-v2 (Recht et al., 2019), ImageNet-Adversarial (Hendrycks et al., 2021b), ImageNet-Retention (Hendrycks et al., 2021a), and ImageNet-Sketch (Wang et al., 2019). These four datasets do not have training images and their categories correspond to ImageNet (Deng et al., 2009). We train on ImageNet and test on these variant datasets to evaluate domain-generalization performance.

Table 2: Few-shot accuracy on ImageNet (%). $n$-shot denotes training with $n$ samples per class. †: Note that the Zero-Shot CLIP uses no training data of ImageNet. We put this result to the column "Full" for easy comparison. Our results are marked in ​gray​. The best results are **bolded**.

| Method | L Encoder | Full | 1-shot | 2-shot | 4-shot | 8-shot | 16-shot |
|---|---|---|---|---|---|---|---|
| Zero-Shot CLIP | ✓ | 58.2† | - | - | - | - | - |
| Linear Prob. CLIP | ✗ | 72.8 | 23.6 | 32.2 | 40.8 | 48.9 | 54.3 |
| CoOp | ✓ | 65.6 | 59.2 | 59.4 | 59.7 | 61.0 | 63.3 |
| CoCoOp | ✓ | 65.1 | 57.4 | 57.8 | 58.2 | 58.5 | 59.0 |
| CLIP-Adapter | ✓ | - | 58.2 | 58.6 | 59.4 | 60.4 | 61.3 |
| Tip-Adapter | ✓ | - | 57.1 | 57.8 | 58.6 | 59.9 | 61.0 |
| DeFo (ours) | ✓ | **73.2** | **59.4** | **59.7** | **60.3** | **61.7** | **64.0** |

### 4.1.3 TECHNICAL DETAILS

The experiments are built upon CLIP pretrained models. During training, the weights of both image and text encoders are frozen. In this paper, we explore both few-shot and full-dataset training. The few-shot setting follows CLIP (Radford et al., 2021) and CoOp (Zhou et al., 2021), i.e., training with 1, 2, 4, 8, and 16 samples per class that are randomly selected from the training set. By default, we use simple data augmentation of random crop and flip, and train with a SGD optimizer with a mini-batch size of 32, 2e-3 learning rate, 0.9 momentum, and 0.01 weight decay (following CoOp (Zhou et al., 2021)) for 50 epochs. For full-dataset training on ImageNet, we use a batch size of 256 and a learning rate of 0.01, which yields similar accuracy to the default setting but significantly reduces training time.

The number of text queries ($n$) is naturally fixed to the number of classes ($k$) for zero-shot CLIP, CoOp and Target Optimization. We set the length of learnable prompt to 16 words for CoOp, and set the length of learnable class name to two words for Target Optimization. The query size of DeFo is scalable in terms of both the length and quantity of text, so we have flexible choices. We empirically find that a larger query size (the number of text queries $n$) generally yields better predictive performance, in particular for large-scale datasets such as ImageNet (Deng et al., 2009). For example, with a similar number of text queries, i.e., $n = 1000$ for CLIP and $n = 1024$ for DeFo, DeFo outperforms the zero-shot CLIP by 14.1% (top-1 acc.) on ImageNet, while this improvement can be further boosted to 15.0% by using 2048 queries in DeFo.

When training on full ImageNet, we use $n = 2048$ text queries and $m = 16$ words (following CoOp (Zhou et al., 2021)) to fully exploit its learning capacity. For few-shot training on ImageNet, we use a smaller query size, i.e., $n = 1024$ and $m = 4$, to prevent over-fitting. For the other 10 datasets, the text length is set to 16, and we find that a smaller number of queries could be sufficient to yield good performance. Specifically, considering the scale of each dataset, we set $n = 1024$ for SUN397 (Xiao et al., 2010), $n = 512$ for StanfordCars (Krause et al., 2013), Food101 (Bossard et al., 2014), and UCF101 (Soomro et al., 2012), $n = 256$ for Caltech101 (Fei-Fei et al., 2004), Flowers102 (Nilsback & Zisserman, 2008), and FGVCAircraft (Maji et al., 2013), and $n = 128$ for OxfordPets (Parkhi et al., 2012), DTD (Cimpoi et al., 2014), and EuroSAT (Helber et al., 2019).

For CoOp, we follow its default setup to initialize the trainable text embeddings from randomness, as we find that the random initialization and manual initialization (e.g., initialize from "a photo of a") yield almost the same performance. When training on full datasets, this phenomenon also works for DeFo and Target Optimization, so we randomly initialize the parameters as well. For few-shot training of DeFo, we initialize the first $k$ text queries by the $k$ class names with random prefix, and fix the corresponding weights ($W \in \mathbb{R}^{k \times k}$) of the classification layer to an identity matrix. In this way we further reduce the number of trainable parameters and make use of language supervision via the text encoder, which consequently yields robust performance when training data is limited.

## 4.2 MAIN RESULTS

### 4.2.1 COMPARISON ON IMAGENET

We first compare our DeFo with the baselines on ImageNet under both full-dataset training and few-shot settings. As shown in Table 1, by training on the entire ImageNet data, our method ob-

Table 3: Domain transfer accuracy on ImageNet variants (%). Our results are marked in gray .

| Method | L Encoder | ImageNet-v2 | ImageNet-A | ImageNet-R | ImageNet-S |
|---|---|---|---|---|---|
| Zero-Shot CLIP | ✓ | 51.5 | 21.7 | 56.0 | 32.9 |
| Lin-Probe CLIP | ✗ | 52.1 (+0.6) | 13.6 (-8.1) | 35.5 (-20.5) | 21.8 (-11.1) |
| CoOp | ✓ | 55.3 (+3.8) | 22.4 (+0.7) | 55.9 (-0.1) | 33.5 (+0.6) |
| DeFo (ours) | ✓ | 58.4 (+6.9) | 21.7 (+0.0) | 55.8 (-0.2) | 33.2 (+0.3) |

Table 4: Average test accuracy (%) on 11 datasets. Results with † are taken from (Gao et al., 2021). Our results are marked in gray . The best results are **bolded**. The results without using text encoder are de-emphasized.

| Method | RN-50 | RN-101 | ViT-B/32 | ViT-B/16 |
|---|---|---|---|---|
| Zero-Shot CLIP (Radford et al., 2021) | 58.9 | 59.9 | 61.6 | 65.2 |
| Linear-Probing CLIP | 79.2 | **81.9** | 74.7 | 80.0 |
| CoOp (Zhou et al., 2021) | 73.7 | 76.2 | 75.5 | 79.7 |
| Target Optimization | 76.1 | 78.2 | 76.3 | 80.8 |
| CLIP-Adapter (Gao et al., 2021) | 74.6† | - | - | - |
| DeFo (ours) | **79.9** | **82.5** | **80.8** | **82.8** |

tains the highest test accuracy with both ResNet and Vision Transformer backbones. Notably, with a ResNet-50 image encoder, our DeFo outperforms the zero-shot CLIP by 15.0%. It is also observed that by using better prompts (i.e., Prompt Ensembling and CoOp), the accuracy is improved by a relatively small margin. This result demonstrates the issues of **expressive sensitivity**, i.e., the human-annotated class names cannot define or well describe the semantic information of the images in each category, even if the prompt has been optimized. Notably, using a simple prompt but optimizing the class names (Target Optimization) yields more competitive performance (e.g., 71.4% vs. 65.6%).

Overall, our DeFo continues to yield superior performance than the baselines for both full-dataset and few-shot training as shown in Table 2. The linear-probing protocol achieves close accuracy to DeFo with sufficient training samples (e.g., 72.8% vs. 73.2%). However, its drawback is obvious when training data is limited. Typically, as reported in Table 2, the linear probing protocol with one sample per class yields only 23.6% accuracy, which is much lower than that of zero-shot CLIP (58.2%), CoOp (59.2%), and our DeFo (59.4%).

### 4.2.2 GENERALIZED PERFORMANCE

We evaluate the domain-transfer performance by 16-shot training on ImageNet and testing on ImageNet-v2 (Recht et al., 2019), ImageNet-Adversarial (Hendrycks et al., 2021b), ImageNet-Retention (Hendrycks et al., 2021a), and ImageNet-Sketch (Wang et al., 2019). As shown in Table 3, compared with the baseline of zero-shot CLIP, DeFo attains 6.9% higher accuracy on ImageNet-v2. Also, DeFo yields a similar level of transfer performance as zero-shot CLIP and CoOp did on the other three datasets. In contrast, the linear probing protocol incurs significantly degraded performance on ImageNet-A, -R, and -S, as it forgoes assistance of language information.

For a wider range of classification tasks, we further evaluate DeFo on a total of 11 datasets. As shown in Table 4, our DeFo achieves the highest average test accuracy over the 11 benchmarks with different image encoders. A specific comparison to CLIP and CoOp on each of the datasets is also provided in Figure 2. We note that CLIP favors the common and generic objects such as the images in Food101, OxfordPets, and Caltech101, for which our DeFo outperforms CLIP by < 10% accuracy and CoOp even fails to improve upon CLIP on Food101. However, when it comes to fine-grained feature recognition tasks such as classifying the type of aircraft (Maji et al., 2013), CLIP and CoOp are shown to be very sensitive to the objects. Consequently, DeFo outperforms CLIP by 25.4% accuracy and outperforms CoOp by 11.2% on this dataset. The different robustness between CLIP and DeFo on the 11 datasets indicates the issue of **sensitivity challenge** for CLIP, and indicates that DeFo successfully addresses this issue by decomposing and then combining the visual features.

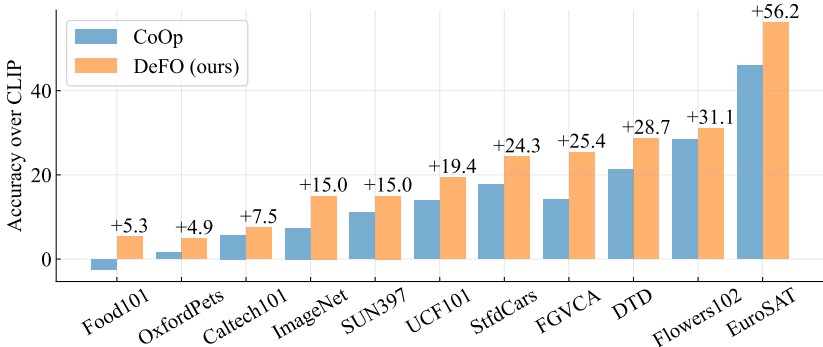

Figure 2: Accuracy improvements over zero-shot CLIP. On all the 11 classification benchmarks, our method outperforms the CLIP and CoOp baselines by non-trivial margins.

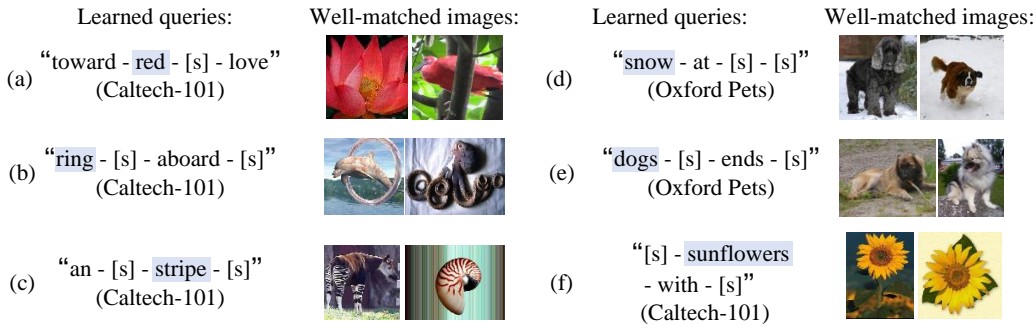

Figure 3: Interpretation (nearest words) of the learned text embeddings of DeFo. We highlight the key words and replace the symbols and meaningless words by "[s]". We surprisingly find that our DeFo is able to learn detailed visual features such as color (a), shape (b), texture (c), and context (d). Also, DeFo is able to directly learn a precise semantic target (f, sunflower is a category of Caltech-101) or a generalized semantic target (e).

## 4.3 INTERPRETATION OF TEXT QUERIES

One benefit of CLIP-like models is that they are able to provide interpretable visual predictions, as the visual features are highly aligned with the representations of natural language. A simple way to interpret the learned word embeddings, i.e., the $n$ sequences of $m$ embedding vectors in $\mathbf{X}_L$, is searching the nearest natural words within the vocabulary by measuring their Euclidean distance. However, as this approach directly maps the continuous embedding vectors into discrete codes of words, the interpreted sentences do not necessarily "make sense" and may contain meaningless words or symbols, which is also observed in prior work (Zhou et al., 2021).

Nonetheless, we still find very interesting evidence from the interpretation of DeFo. We observe that some of the interpreted query sentences include meaningful key words that describe specific visual features such as color, shape, and texture. As illustrated in Figure 3 (a)-(c), in Caltech-101 dataset (Fei-Fei et al., 2004), DeFo learns the words "red", "ring", and "stripe", while the well-matched (based on the consine similarity in the latent space) images in the dataset look consistent with human understanding of these features. For example, DeFo matches the word "red" with the objects such as a lotus flower and a bird in this color. For the word "ring", we can find the ring or circle shapes in the corresponded images. Also, DeFo is able to extract background information such as "snow" (see Figure 3 (d)). And surprisingly, DeFo sometimes directly learns the semantic targets that are closely related to the categories of the dataset. For example, it learns the word "dogs" which is a parent category in OxfordPets (Parkhi et al., 2012), and the word "sunflowers" which is an exact category in Caltech-101 (Fei-Fei et al., 2004).

Despite the fact that this interpretation approach is not rigorous enough, because the text features learned by DeFo possibly exceed the existing vocabulary, it still provides very strong evidence that DeFo features are meaningful. We hope this result will yield greater insights in a follow-up study on interpretable vision-language inference.

### 4.4 ABLATION STUDY

In this section, we present additional results to ablate the gains of DeFo. First, the size of the textual input, including the number of queries $n$ and the length $m$ of each query sentence, may affect the performance. We compare the accuracy of a DeFo model with different $n$ and $m$ and summarize the results in Table 5a and 5b, where we use a ResNet-50 image encoder and train our model on the entire ImageNet data. As reported in the two tables, a smaller size of textual input slightly reduces the performance within an acceptable margin. For example, with $n = 256$ queries it yields 72.5% accuracy on ImageNet, which is only 0.7% lower than that of the default setup of $n = 2048$. Also, using $m = 4$ words per query yields only 0.5% lower accuracy than that of $m = 16$, and further increasing $m$ (e.g., $m = 32$) cannot obtain clear improvements. These results indicate that DeFo is robust to its hyper-parameters, and DeFo improvements of accuracy do not rely on large-scale trainable parameters.

There is another concern that DeFo uses an additional classification layer which introduces $n \times k$ more parameters, where $n$ and $k$ denote the number of queries and classes, respectively. To ablate the gain of the additional parameters, we add the same layer on top of CLIP and CoOp, and compare their performance with DeFo. Specifically, we attach a $k \times k$-dimensional linear layer to the logits of CLIP and CoOp, so their model size is identical to DeFo if we set $n = k$. We conduct this experiment on ImageNet as well, and the results are summarized in Table 5c. As is shown, with the help of the linear layer, the accuracy of CLIP is improved to 62.4%, which is still significantly lower than that of DeFo. Notably, the "CLIP + linear" model is equivalent to our DeFo model with $n = k$ and fixing the textual inputs to the queries used in CLIP. This indicates that the classification layer yields a very limited improvement ($< 4\%$), and the superior performance of DeFo mainly comes from it learning decomposed features.

Table 5: Ablation studies of DeFo. Our results with default setup are marked in gray .

(a) Accuracy with $m$=16.

| Queries ($n$) | Acc. |
| --- | --- |
| 256 | 72.3 |
| 512 | 72.5 |
| 1024 | 72.9 |
| 2048 | 73.2 |

(b) Accuracy with $n$=2048.

| Length ($m$) | Acc. |
| --- | --- |
| 2 | 72.5 |
| 4 | 72.7 |
| 8 | 73.0 |
| 16 | 73.2 |
| 32 | 73.2 |

(c) Gain of linear layer.

| Model | Acc. |
| --- | --- |
| CLIP | 58.2 |
| CLIP + linear | 62.0 |
| CoOp | 65.6 |
| CoOp + linear | 69.8 |
| DeFo ($n$=1000) | 72.9 |

## 5 CONCLUSION

In this paper, we identify two main issues of existing vision-language inference protocols, i.e., the expressive sensitivity and the conceptual sensitivity. To address them, we propose DeFo which maintains the dual-model architecture but infers by decomposed visual features. Specifically, it leverages trainable text prompts and decouples visual features from hard semantic targets. We demonstrate the significance of DeFo by showing its two benefits. First, DeFo gets rid of the textual descriptions of class names and instead infers via a linear classifier, which yields superior performance in the full-dataset scenarios compared with zero-shot CLIP and CoOp. Next, DeFo keeps the language encoder, which we find is able to bound the projection of visual features and therefore achieves competitive results in few-shot learning and domain transfer. Overall, DeFo provides a new vision-language learning and inference paradigm, i.e., prompting the decomposed visual features, which we hope is of practical importance in fully exploiting the learning capacity of vision-language models.

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
