# OpenReview forum: "Learning to Decompose Visual Features with Latent Textual Prompts"
_ICLR.cc/2023/Conference — ICLR 2023 poster_

### Official Review · Reviewer_xRfX · 2022-10-22

**Confidence:** 3
**Correctness:** 4
**Technical Novelty And Significance:** 4
**Empirical Novelty And Significance:** 4
**Recommendation:** 8

**Clarity, Quality, Novelty And Reproducibility:**

Overall, this paper is well written and the method is easy to follow. The motivation is well explained and the contributions is novel.

**Strength And Weaknesses:**

Strengths:
1）Prompts Learning is a meaningful direction, and this paper provides a valuable discussion of this direction to some extent.
2) The authors propose a novel and efficient method to address two issues regarding the Clip-like model.
3) The DeFo achieves 73.2% test accuracy on ImageNet with a ResNet-50 backbone without tuning any pretrained weights of both the vision and language encoder, outperforming zero-shot CLIP by a large margin of 15.0%, and outperforming state-of-the-art vision-language prompt tuning by 7.6%.
4）This paper is easy to follow and the motivation is well explained.

Weaknesses:
There are some issues that need to be improved: 1) The introduction and related work section lack a detailed discussion of the recent method CoCoOp. 2) A comparison with the SOTA method CoCoOp is lacking in the experimental section. For example, in Tab1, Tab2, Tab3, Tab4 and Fig2.


**Summary Of The Paper:**

This paper aims to address two issues regarding the Clip-like model: 1) degraded accuracy and robustness when inferring by retrieving textual class names (the zero-shot protocol); 2) breaking the well-established vision-language alignment (linear probing). To combine the best of both worlds, this paper proposes Decomposed Feature Prompting (DeFo), which maintains the dual-model architecture yet leverages learnable embeddings as textual input and performs classification with an additional linear layer. The empirical study shows DeFo’s performance in improving the vision-language models.

**Summary Of The Review:**

In general, this paper is well written. I suggest that this paper can be accepted after supplementing some contrastive results against SOTA method and more discussions in the experimental part.

---

> ### Author Response · Authors · 2022-11-17
> **Response to Reviewer xRfX**
>
> We appreciate your careful review and feedback on this paper. Below are our detailed responses.
>
> Q1: "The introduction and related work section lack a detailed discussion of the recent method CoCoOp; A comparison with the SOTA method CoCoOp is lacking"\
> A1: Thanks for this suggestion. We report the comparison to CoCoOp in the table below, in which DeFo yields better results. Note that CoCoOp is an extension of CoOp which aims to address a different problem than the one we target in this paper. Specifically, CoCoOp focuses on transferring CoOp’s learned prompt prefix from a portion of classes (source) to unseen classes (target). This leads to worse results than CoOp in the source classes due to the trade-off between the source and target performance. It’s a great suggestion to provide more detailed discussions in our paper, and we have highlighted this point and included a detailed comaprison to CoCoOp in the revised version.
> | Method      | Full-ImageNet | 16-shot ImageNet | ImageNet variants | Avg. score in 11 datasets |
> | ----------- | ------------- | ---------------- | ----------------- | ------------------------- |
> | CLIP        | 58.2          | -                | 40.5              | 58.9                      |
> | CoOp        | 65.6          | 63.3             | 41.8              | 73.7                      |
> | CoCoOp      | 65.1          | 62.7             | 41.7              | 72.9                      |
> | DeFo (ours) | 73.2          | 64.0             | 42.3              | 79.9                      |

---

> > ### Comment · Reviewer_xRfX · 2022-11-24
> > **Thanks for the reply**
> >
> > I appreciate the effort  of the authors in supplementing contrastive experiments. The experimental results and discussion are convincing. I will keep my original rating.

---

> > > ### Author Response · Authors · 2022-11-24
> > > **Thanks for your reply**
> > >
> > > We are very happy that the response helps addressing your concerns. Thanks again for your constructive review on the paper!

---

### Official Review · Reviewer_gg5B · 2022-10-24

**Confidence:** 3
**Correctness:** 4
**Technical Novelty And Significance:** 2
**Empirical Novelty And Significance:** 3
**Recommendation:** 6

**Clarity, Quality, Novelty And Reproducibility:**

Good clarity. I have no comments on reproducibility as source codes are not released.

**Strength And Weaknesses:**

Strength:
1. Good and clear motivation, good paper writing with easy understanding scope&contribution.
2. Great empirical results.


Weaknesses:
1. Limitation of technical novelty but it may not be a big concern from my perspective.

**Summary Of The Paper:**

This paper presents a simple method of prompt learning for Vision-Langauge model (in particular the CLIP). It is an extension based on existing work CoOp, where it leverages learnable embeddings as textual input and performs classification on this basis. Empirically, DoFo shows great improvement, where it reaches 73.2% on ImageNet 1K benchmark. In overall, it is a simple yet effective way for the VL prompt learning.

**Summary Of The Review:**

In overall this is a good extension on the previous CoOp, where the authors aim to improve the prompt learning for a pre-trained VL model, in particular the CLIP. The proposed method is simple: by leveraging set of learnable parameters as textual embedding, it assist the classification task especially on ImageNet chanllenge. In essense, DeFo is largely based on the exisitng technique of CoOp, yet it's presented empirical results largely exceed the previous. My biggest concern arises from the aspect of technical contribution and the larger impact of CoOp alike work:

1. The reviewer understands that simple technique is possibly to lead to bigger impact, so it is indeed not an issue for a intuitive&direct solution as a good submission. Yet my hesitation comes from the consideration that the presented insight of DeFo remains less significant: it seems that the biggest modification is making the learnable embedding as input for textual encoder yet it seems the important baseline ``Target Optimization'' also reaches a good results in such a similar schema. But again, I understand and also agree that the complexity of technique does not reflect contribution so I prone to suggest acceptance for the submission.

2. Impact of CoOp alike work: another of my concern comes from the real world impact of prompt learning and its variations in recent years. Indeed, prompt learning, especially for LLMs show promissing applications for language comprehending tasks, yet this seems to be less important on the Vision side, which is mainly because that the pre-trained Vision model is that large enough that requires prompt learning. E.g., tiny model like MobileNet can even perform better than CLIP or other Vision tasks much more efficiently and better. A more desperate need of prompt engineering is for the Vision-Language Tasks that require more comprehending and high-level understanding, e.g., image/video question answering, captioning and cross-modal retrieval. In such context, prompt engineering for CLIP is rather limited and does not reflect much extension yet as current efforts seem more focusing on ImageNet challenge for simple classification task. But admittedly, the reviewer still agree that DeFo is in overall a good submission and has it's contribution for the community.

---

> ### Author Response · Authors · 2022-11-17
> **Response to Reviewer gg5B**
>
> We appreciate your careful comments and constructive suggestions. The followings are detailed responses to your questions/concerns:
>
> Q1: "My hesitation comes from the consideration that the presented insight of DeFo remains less significant: it seems that the biggest modification is making the learnable embedding as input for textual encoder yet it seems the important baseline Target Optimization also reaches a good results in such a similar schema."\
> A1: Thanks for your comments. Our DeFo aims to address the challenges of expressive sensitivity and conceptual sensitivity, while maintaining the vision-language dual architecture. For this, we 1) use learnable embeddings instead of manually given class name words (as pointed out by the reviewer); and we 2) map the image features into a latent space for vision-language alignment and perform classification via a *linear projection* afterwards instead of using a direct retrieval.\
> &emsp; Importantly, note that the “Target Optimization” baseline is proposed in this DeFo paper as an ablation of DeFo to understand the importance of the learnable embeddings (point 1). For example, with a ResNet-50 image encoder, CoOp obtains an average score of 73.7% over 11 datasets, while Target Opt. (use of learnable embeddings) improves this number by 2.4%, and DeFo (use of learnable embeddings and linear projection) further improves Target Opt. by 3.8%. This suggests that both of our contributions help to improve the results.
>
> Q2: "Another of my concern comes from the real world impact of prompt learning and its variations in recent years."\
> A2: Thank you for sharing your thoughts on prompt learning. Prompt learning is indeed an effective technique for language models with promising real-world applications. We also believe prompt learning has great potential in vision-language understanding. Despite the fact that recent advances of vision-language prompt engineering still focus on simple classification tasks, we go one step beyond and investigate not only the classification performance but also identify decomposed visual features that can be learned by prompting. For example, we find some interesting evidence of the visual features that the model focuses on (see Fig.3). From our perspective, prompt learning is able to lead to a more intuitive understanding of visual representations. We hope this work provides valuable insights for follow-up studies in vision-language prompt learning.

---

> > ### Comment · Reviewer_gg5B · 2022-11-23
> > **Thanks for the response**
> >
> > Thanks the author for their response and I'd like to keep my initial rating for the submission.
> >
> > Regards

---

> > > ### Author Response · Authors · 2022-11-24
> > > **Thanks for your reply**
> > >
> > > Thank you again for your constructive comments and suggestions to the paper. And please feel free to leave any new comments if you have further questions/concerns. We are happy to discuss them with you.
> > >
> > > Thanks

---

### Official Review · Reviewer_MDxU · 2022-10-27

**Confidence:** 3
**Clarity, Quality, Novelty And Reproducibility:** The paper is well written and easy to…
**Correctness:** 3
**Technical Novelty And Significance:** 3
**Empirical Novelty And Significance:** 3
**Recommendation:** 5

**Strength And Weaknesses:**

Strength：
1. The two problems pointed out by the paper: do exist and deserve the attention of the community.
2. The authors take time to implement and evaluate several prominent baselines. Experimental evaluation shows competitive performance.
3. This paper is well written and easy to follow.

Weakness:
1. The paper does not discuss the computational complexity of the proposed methods. How to efficiently complete the fine-tuning of the pre-trained model is also a direction worthy of attention. I look forward to seeing the authors discuss a comprehensive comparison of DeFo's training time and other methods, such as CoOp and CLIP-adapter.
2. Some important ablation study may be missing.  Since the authors point out that using class labels to generate text embeddings may bring challenges with expressive sensitivity. So a very straightforward idea is that we can directly set an independently learnable parameter as the prototype of each category to calculate the cosine similarity with image embeddings. or adopt the exponential-moving-average (EMA) manner [3]. These above-mentioned methods do not use text encode. Therefore, it is not necessary to carry out the forward of the text encoder every iteration during training. If these method can also achieve very good results, then I feel that the novelty and effectiveness of DeFo may be challenged. Therefore, I think it is very necessary to supplement this experiment. Look forward to the author discussing in following version.
3. The comparison of some other important baseline is missing, such as Tip-adapter [1] and CoCoOp [2].
4. I look forward to the author's discussion of the additional learnable parameters introduced in addition to CLIP's pre-trained model, and compare the number with other methods. Because if too many parameters are introduced, the performance improvement may come from overfitting of too many parameters. If the authors would like to compare the number of additional parameters of DeFo with CoOp and CLIP-adapter, I think it may be very helpful for us to comprehensively evaluate and compare these methods.
5. We expect that the model can not only achieve good performance on a single dataset, but also have the potential to transfer beyond a single dataset. I suggest authors to add discussion about the perfomance of DeFo for domain generalization. Maybe the setting in Section 4.2 of [3] is a good formulation. This may strengthen the contribution of the paper.


**Summary Of The Paper:**

This paper point out two challenges in the downstream inference of pre-training vision-language models: expressive sensitivity and conceptual sensitivity.  To handle the problems, the paper proposes a new dual-model feature prompting methd, named as Decomposed Feature Prompting (DeFo). By providing an independent set of learnable embedding and tuning an additional layer for classification, authors claim that the model trained by DeFo significantly addresses the sensitivity challenges of CLIP-like models.


**Summary Of The Review:**

The proposed approach is shown to be effective, but the lack of some experiments may lead to the limited contribution of the proposed method. If the author adds more meaningful experiments, I think it will make the paper more interesting, and I am very happy to revise my score.

---

> ### Author Response · Authors · 2022-11-17
> **Response to Reviewer MDxU (part 1)**
>
> Thank you very much for your comments and suggestions to this paper. We address your concerns as below.
>
> Q1: The discussion of computational complexity a comprehensive comparison of training time.\
> A1: Great question: Overall, the training time per batch between these methods doesn’t differ much as shown in the table below. The differences can be explained by looking at the additionally introduced computations. DeFo introduces only one additional linear layer with n × k parameters (n and k denote the number of queries and classes) compared to CoOp. In contrast, CoCoOp introduces two more layers and CLIP-Adapter introduces four more layers compared to CoOp. In summary, we find DeFo to yield much better results while having a better training time than CoCoOp and CLIP-Adapter.
> | Method         | DeFo  | CoOp  | CLIP-Adapter | CoCoOp |
> | -------------- | ----- | ----- | ------------ | ------ |
> | batch time (s) | 0.442 | 0.428 | 0.457        | 0.446  |
>
> Q2: "Some important ablation studies may be missing. A very straightforward idea is that we can directly set an independently learnable parameter as the prototype of each category to calculate the cosine similarity with image embeddings. or adopt the exponential-moving-average (EMA) manner."
> A2: Thanks for pointing out this baseline of training prototype vectors without using a text encoder. Note that this is equivalent to linear probing: we train a d × k dimensional matrix (d is the dimension of latent feature and k is the number of classes) and each d-dimensional vector in this matrix can be considered as an independently learned prototype for a category. As discussed in the paper (see Section 4.2 for details), in the absence of a text encoder, this form of linear probing requires sufficient downstream training samples to learn domain-adapted features, i.e., it yields poor few-shot and domain-generalized performance. The related results can be found in Table 2 and Table 3. For example, while the quantity of training samples is limited, linear probing yields much lower accuracy than our DeFo (e.g., 40.8 vs. 60.3, 4-shot), and when transferring the learned features from the original ImageNet to its variants such as ImageNet-Adversarial, ImageNet-Retention, and ImageNet-Sketch, linear probing also suffers from a severe degradation of accuracy (e.g., 35.5 in ImageNet-R vs. 55.8). Also, as linear-probing freezes the parameters of the encoder and optimizes only the last linear layer, employing EMA is not expected to make significant improvements. We test this protocol in ImageNet with CLIP’s ResNet-50 image encoder and an accuracy of 72.83% is observed, which is only 0.05% higher than that of linear-probing without EMA.
>
> Q3: "The comparison of some other important baseline is missing, such as Tip-adapter and CoCoOp."\
> A3: Thanks for the suggestion. We compare to Tip-Adapter and CoCoOp in the table below and observe that our DeFo yields significantly better results. More detailed comparisons to these two baselines have also been included in our revised paper. Note, we expect CoCoOp and Tip-Adapter to yield worse results than their base models CoOp and CLIP-Adapter because they address a different problem. Specifically, while our DeFo focuses on addressing CLIP’s sensitivity challenges and improving its downstream performance, CoCoOp focuses on transferring CoOp’s learned prompt prefix from a portion of classes (source) to unseen classes (target). This leads to worse results for the source classes because of the trade-off between the source and target performance. Similarly, Tip-Adapter focuses on  saving CLIP-Adapter’s training time by conducting a non-parametric layer with the representation of few-shot prototypes, which however worsens the model’s scalability and leads to relatively lower results because the non-parametric layer is very sensitive to the random chosen of few-shot prototypes.
> | Method         | Full-ImageNet | 16-shot ImageNet | ImageNet variants | Avg. score in 11 datasets |
> | -------------- | ------------- | ---------------- | ----------------- | ------------------------- |
> | CLIP | 58.2          | -                | 40.5              | 58.9                      |
> | CLIP-Adapter   | -             | 63.6             | -                 | -                         |
> | Tip-Adapter    | -             | 62.0             | -                 | -                         |
> | CoOp           | 65.6          | 63.3             | 41.8              | 73.7                      |
> | CoCoOp         | 65.1          | 62.7             | 41.7              | 72.9                      |
> | DeFo (ours)    | 73.2          | 64.0             | 42.3              | 79.9                      |

---

> ### Author Response · Authors · 2022-11-17
> **Response to Reviewer MDxU (part 2)**
>
> Q4: "I look forward to the author's discussion of the additional learnable parameters introduced. If too many parameters are introduced, the performance improvement may come from overfitting."\
> A4: Thanks for the suggestion. We have added a discussion of learnable parameters in the revised version. The additionally trainable parameters of DeFo consist of 1) trainable text embeddings, and 2) the projection layer. Specifically, DeFo has N=n×m×d+n×k additionally trainable parameters, where n denotes the number of text queries, m refers to the length of each query sentence, d is the dimension of the text embeddings, and k is the number of classes. Thus, for ImageNet (k=1000), with m=16, n=1024, d=512 (identical to CoOp), DeFo uses N=9.4M additional parameters. Similarly, CoOp has N=8.2M additional trainable parameters for class specific prompts. CLIP-Adapter uses N=1M additional parameters.\
> &emsp;Notably, we don’t observe any overfitting because DeFo has good domain-generalization performance (see Table 3), and demonstrates to be robust to the size of trainable parameters (see Table 5). Specifically, in Table 5(a), when reducing n from 1024 to 256 (i.e., the size of DeFo’s trainable parameters is four times smaller), DeFo’s accuracy drops by only 0.6%. Similarly, in Table 5(b), when reducing m from 32 to 2, the accuracy drops by 0.7%. With a smaller model size (n=256, m=16), our DeFo (72.3%) still significantly outperforms CoOp’s 65.6% (n=1000, m=16) and CLIP-Adapter’s 61.3%. Meanwhile, this DeFo model introduces N=2.4M learnable parameters, less than CoOp’s N=8.2M for class specific prompts.
>
> Q5: "We expect that the model can also have the potential to transfer beyond a single dataset. I suggest authors to add discussion about the performance of DeFo for domain generalization. Maybe the setting in Section 4.2 of [3] is a good formulation."\
> A5: Thanks for the suggestion. We are not entirely sure about what [3] refers to, and assume that the DINO paper was intended. Please let us know if we misunderstood. Following Section 4.2 of DINO, we explore the performance in retrieval of off-the-shelf features in the “revisited Oxford and Paris” image retrieval dataset. Following DINO, we report the mAP scores for the Medium (M) and Hard (H) splits as below (with ResNet-50 image encoder). We find DeFo to slightly outperform the original CLIP model. Despite better performance, DeFo hence also generalizes equally well. This result has been included into the supplementrary material.
> |      | R-Ox (M) | R-Ox (H) | R-Par (M) | R-Par (H) |
> | ---- | -------- | -------- | --------- | --------- |
> | CLIP | 36.1     | 11.8     | 56.4      | 27.9      |
> | DeFo | 36.3     | 11.8     | 56.6      | 58.0      |

---

> ### Author Response · Authors · 2022-12-11
> **Post-Rebuttal Discussions**
>
> Dear reviewer MDxU,
>
> Thank you again for reviewing our manuscript and giving constructive comments. As the deadline of discussion is quite near, we are wondering if our response has addressed your concerns to the paper? Your feedback will be invaluable in improving the paper and any new comments will be greatly appreciated.

---

### Decision · Program_Chairs · 2023-01-20

**Decision:**

Accept: poster

**Justification For Why Not Higher Score:**

interesting and solid work but not particularly novel

**Justification For Why Not Lower Score:**

N/A

**Metareview: Summary, Strengths And Weaknesses:**

The paper proposed a new vision-language prompting method called dual-model feature prompting to address two issues of the Clip-like models: 1) degraded accuracy and robustness when inferring by retrieving textual class names (the zero-shot protocol); 2) breaking the well-established vision-language alignment (linear probing), and empirically validates DeFo’s performance in improving the vision-language models. Reviewers generally found this work is novel, well-motivated, empirical results are solid and the paper is well-written. There are some concerns raised by the reviewers and authors provided a proper response. Overall, this is an interesting paper with solid contribution which is recommended to publish.



**Note From Pc:**

if the above contains the word "oral" or "spotlight" please see: "oral" presentation means -> notable-top-5% and "spotlight" means -> notable-top-25%. As stated in our emails, we are disassociating presentation type from AC recommendations